# Design and Evaluation of a Button Sensor Antenna for On-Body Monitoring Activity in Healthcare Applications

**DOI:** 10.3390/mi13030475

**Published:** 2022-03-20

**Authors:** Shahid Muhammad Ali, Cheab Sovuthy, Sima Noghanian, Tale Saeidi, Muhammad Faran Majeed, Amir Hussain, Faisal Masood, Shariq Mahmood Khan, Syed Aziz Shah, Qammer H. Abbasi

**Affiliations:** 1Department of Electrical and Electronic Engineering, Universiti Teknologi, PETRONAS Bander Seri Iskandar, Tronoh 32610, Perak, Malaysia; shahid_17006402@utp.edu.my (S.M.A.); faisal_17001061@utp.edu.my (F.M.); 2CommScope Ruckus Wireless, 350 W Java Dr, Sunnyvale, CA 94089, USA; 3POSTECH # 413 LG Research Bldg., # 77 Cheongam-ro, Pohang-si 37673, Korea; talecommunication@gmail.com; 4Department of Computer Science, Kohsar University, Murree 47150, Pakistan; m.faran.majeed@ieee.org; 5School of Computing, Edinburgh Napier University, Edinburgh EH11 4BN, UK; a.hussain@napier.ac.uk; 6Department of Computer Science & IT, NED University of Engineering & Technology, Karachi 75270, Pakistan; shariq@neduet.edu.pk; 7Research Centre for Intelligent Healthcare, Coventry University, Coventry CV1 5RW, UK; syed.shah@coventry.ac.uk; 8James Watt School of Engineering, University of Glasgow, Glasgow G12 8QQ, UK; qammer.abbasi@glasgow.ac.uk

**Keywords:** button sensor antenna, body centric communication, specific absorption rate (SAR)

## Abstract

A button sensor antenna for on-body monitoring in wireless body area network (WBAN) systems is presented. Due to the close coupling between the sensor antenna and the human body, it is highly challenging to design sensor antenna devices. In this paper, a mechanically robust system is proposed that integrates a dual-band button antenna with a wireless sensor module designed on a printed circuit board (PCB). The system features a small footprint and has good radiation characteristics and efficiency. This was fabricated, and the measured and simulated results are in good agreement. The design offers a wide range of omnidirectional radiation patterns in free space, with a reflection coefficient (S_11_) of −29.30 (−30.97) dB, a maximum gain of 1.75 (5.65) dBi, and radiation efficiency of 71.91 (92.51)% in the lower and upper bands, respectively. S_11_ reaches −23.07 (−27.07) dB and −30.76 (−31.12) dB, respectively, with a gain of 2.09 (6.70) dBi and 2.16 (5.67) dBi, and radiation efficiency of 65.12 (81.63)% and 75.00 (85.00)%, when located on the body for the lower and upper bands, respectively. The performance is minimally affected by bending, movement, and fabrication tolerances. The specific absorption rate (SAR) values are below the regulatory limitations for the spatial average over 1 g (1.6 W/Kg) and 10 g of tissues (2.0 W/Kg). For both indoor and outdoor conditions, experimental results of the range tests confirm the coverage of up to 40 m.

## 1. Introduction

Wireless body area network (WBAN) applications are paving the way for a future in which doctors can have patient-specific information at their fingertips whenever they need it (Figure 1). Wearable devices have recently attracted the interest of researchers [1,2]. However, body presence, robustness, as well as material choice, form factor, mechanical flexibility, and stretchability, all influence wearable devices design [3,4]. Various types of design have been proposed in wearable applications; one of the first sensor antennas was published in [5]. In order to calculate the humidity content of sludge samples, a circular antenna was proposed. The sample was placed in a plastic beaker at the top of the proposed design. Based on the effective dielectric permittivity of the design, the Botcher model was utilized to find out the humidity content. The first sensor antenna dedicated to temperature measurement was reported in [6]. A shape memory polymer (SMP) paper was used to create the sensor, which was sandwiched between a radio frequency identification (RFID) tag and a metallic sheet. The SMP changed its relative permittivity as long as the temperature crossed the threshold, which could be determined by the RFID tag as it was switched on. Sensor antennas, which are fabricated on rigid materials, have been used in a variety of medical and non-medical applications. As an example, [7] describes a microstrip patch antenna-based sensor made on a Flame Retardant 4 (FR4) substrate. This sensor can be used to detect various percentages of sugar and salt by detecting the change in their dielectric properties. Rogers R03006 substrate was used in a microstrip patch antenna to create a sensor described in [8]. The sensor was used to detect the temperature by thermally cycling a patch antenna that was bonded to various metal bases. Rigid substrates are typically unsuitable for wearable applications because they are difficult to bend, stretch or adhere to textile materials [9]. In [10], bio-based waste material was used to create a biodegradable substrate for a microstrip antenna. The antenna was used to show the variation in the dielectric constant of high-permittivity materials such as water and wet soil. Wearable textile sensor antennas have recently garnered a lot of attention due to their ability to detect microstructure deformations and human movements, as well as to monitor a variety of vital signals in healthcare applications [11]. In [12], a wearable finger motion sensor based on a dipole antenna was developed with the dual purpose of detecting and communicating. In addition, the proposed design was attached to a glove to measure the influence of bending the fingers. To monitor human physiology, a planar inverted-F antenna (PIFA) was employed as an on-body wearable device [13]. The design is bulky, and the effect of the surrounding circuit was ignored. Table 1 compares a few of the recent publications related to sensor antennas.

Wearable sensor antennas are often flexible. They are made from flexible materials with a shape that can be used as part of the clothing or garment. Various investigations have focused on the design approach and its realization. Some research activities are focused on the characterization and stability of the substrate as well as sensor antenna performance variations due to the tolerances in their substrate. Other investigations focus on the human body’s influence on the sensor antenna’s performance as well as the safety and power limitations [14,15,16,17,18,19].

Wearable devices have found countless applications and each one of them calls for specific requirements. Wearable technology has been developed in the areas of tattoo and epidermal electronics [20], and flexible and stretchable antennas for bio-integrated electronics and biosensors [21,22,23]. Various technologies and new materials have been developed that enable us to make biodegradable and heterogeneously layered materials to support various microsystem and sensors. While these stretchable, flexible, and integrated biodegradable sensors are remarkable, the cost and requirements might not be feasible for everyday wearable devices. When it is possible to combine a rigid miniaturized antenna with wearable sensors, a more cost-effective and feasible solution might be provided. Wearable sensor antennas should be analyzed in real-world situations, including the existence of the human body, bending and wrinkling of clothing, and washing. The buttons are a good candidate for antennas to be integrated to. They can be easily mounted on textiles materials. Buttons are made of rigid material, which helps in stabilizing the performance of the antenna. Rigid conductors, such as copper, can be used in BSA design. The use of highly conductive metallic vs. e-textile materials that usually have lower conductivity improves the performance of the antenna. Further, the BSA design can be easily integrated into clothes at various positions. While the rigidity of the buttons is an advantage, it is not always possible. The unique design we propose can either be in a button shape or it can be worn using a wristband [24,25].

Our goal was to design a BSA for WBAN applications that would meet the following requirements: (1) multi-band operation for various WBAN bands; (2) small size in the shape of a button to increase the wearer’s comfort (a small rigid PCB size, with an electronic system that could be integrated into a wristband or a jacket’s button); (3) high stability in resonance frequency, impedance matching, and omnidirectional radiation pattern; and (4) good communication range. These properties were not found to be simultaneously satisfied by small sensor antennas proposed in the published literature. The advantage of the proposed BSA is that it does not require special fabrication technology and the material and components are easily available. Further, our proposed BSA is easy to maintain and replace. The standalone unit can be simply detached and replaced, as inspired by [26,27].

In our proposed design, an asymmetrical small-sized rigid PCB is employed as the button, with a flexible conducting textile (clothing) acting as the ground plane while a rigid coaxial feed line is used to feed to a circular sensor module. This BSA provides a small form factor and can easily be integrated into a hand-watch cover. This topology ensures robustness and minimum effects due to bending and stretching. To the best of our knowledge, this is the first time a thorough investigation of a new button sensor antenna system design, and its performance is presented. We propose a complete wireless sensor system that incorporates the newly designed antenna with the transceiver system comprising chips and lumped elements and is equipped to communicate with a WiFi router. This compact sensor is compatible with the IEEE 802.11b/g/n routers.

The paper is arranged as follows: After a brief introduction, Section 2 explains the design requirements and procedure of the wearable button sensor antenna. The measurement results are provided in Section 3. Section 4 concludes the paper with recommendations for future work.

**Table 1 micromachines-13-00475-t001:** Comparison of various miniaturized button-sensor antennas proposed in the literature.

Ref.	Antenna Size (in mm)	Frequency (GHz)	Substrate/Conductor Material	Patterns	Electronic Circuit	Gain (dBi)/Radiation Efficiency	Range	Bending/Wet
[27]	40 × 40 Modular Button	2.40/5.30	Cuming-Foam-PF4/ShieldIt	O	NA	7.80/3.10 (Half-Wave)/NA	NA	NA
[28]	40 × 40 Modular Button	4.75–5.25	Cuming-Foam-PF4/Shieldit	C	NA	NA	NA	NA
[29]	100 × 100 Circular Button	2.40	FR4, Felt /Copper, ShieldIt	O	NA	FS, OA 1.80/5.10, and 97.00/71.00	NA	NA
[30]	18 (Diameter) Button Antenna with Frequency Selective Surface	5.25–5.85	Acrylic Transparent/Copper	B	NA	2.10	NA	NA
[31]	Conical-helix Button-Antenna	3.00/5.80	Copper Surface	O/B	NA	2.30, 8.00/NA		NA
[32]	100 × 100 Circular Button	5.50	FR4, Felt/Copper, ShieldIt (Multilayers)	O	NA	FS, OB 79.90/70.80, 3.50	NA	NA
[33]	80 × 80 Circular Button	2.40/5.00	Rogers 5880, Felt/Copper, ShieldIt	O	NA	FS, OB (Lower/Upper bands): 90/84, 1.05, 0.24/4.50, 4.73	NA	A
[34]	22.3 (Diameter) Circular-Cuff Button	2.40/5.60	PTFE/Copper	O	NA	Free Space 1.50/94.00	NA	NA
[35]	100 × 100 Circular-Array Button	4.50-4.61/ 5.04-5.50	Cuming-Foam (PF4) & RO4003/Copper & ShieldIt	B	NA	7.70/NA	20 mm	NA
[36]	100 × 100 Inverted-F Button	2.45/5.80	RO4003/Copper	O	NA	OB (Lower/Upper bands): −0.60/4.30 and 46.30/69.30	18–122 cm	NA
[37]	Dipole Sensor Antenna	2.40	FR4/Copper	AZ	A	2.40/28.00	1.5 m	NA
[38]	GPS Tracking Antenna	2.40	FR4/Copper	U	NA	3.00/NA	NA	NA
[39]	70 (Diameter) Patch Antenna	2.40/5.80	PDMS/ShieldIt	M	NA	OB (Lower/Upper bands): 4.16/4.34, 58.6	NA	NA
[40]	50 × 57 Sleeve-Badge Antenna	2.45	Multilayered Textile/ShieldIt	O	NA	−2.33/55.3	NA	NA
[41]	130 × 130 Circular Patch with Soft Periodic Surface	4.00	Felt/Shieldex Zell RS	B	NA	NA	NA	A/NA
[42]	70 × 25 Loop Dipole	2.36–2.39	PDMS/Copper	O	NA	FS (OB) 0.68 (−0.40/15.00)	NA	NA
[43]	37.20 × 50 MMA	2.2 to 3.0/5.4 to 6.0	Felt/ShieldIt, Copper Foil	O	NA	OB Chect and OA (L/U) 3.80 (4.67) and 3.00 (4.55)/ 54.37 (54.27), and 60.0 (63.73)	11 m	A
[44]	94.58 × 52.36 Patch	1.80–2.40	Felt/ShieldI	D	NA	NA/50.00	NA	NA
[45]	50 × 50 Graphene Antenna	3.13−4.42	Graphene/Poly.t	O	A	NA	NA	NA
[46]	50 × 60 Patch	2.00–4.00	Cellulose Filter paper	NA	NA	NA	NA	NA
[47]	71 × 64 Patch	2.45	Cotton Jeans	NA	NA	NA	NA	NA
[48]	40 × 40 Reconfigurable Button Antenna	2.45/5.80	PDMS, textile/Copper	B	A	0.70, −0.90, and 43.80/73.80	NA	NA
Pro	Button Antenna: 45 (Diameter) Dual-band Sensor Module	2.45/5.6	PCB Button, Felt/Shield (Substrate/Ground), PCB Module.	O/B	A	(Lower/Upper bands): 2.09 (6.70), 2.160 (5.670)/65.12 (81.63), 75 (85.0) (On Body/Arm)	40 m	A

FS: Free space, C: Circular, OB: On-body, OA: On-arm, NA: Not available, A: Available, O: Omnidirectional, B: Broadside, AZ: Azimuth, U: Unidirectional, and M: Monopole, L/U: Lower/Upper Band, FFS: Frequency Selective Surface, and Rag: Range.

## 2. System Requirement

This section describes the system requirements and design choices. The button antenna and backbone electronics are both investigated. In addition, an appropriate design process is chosen.

### 2.1. System Description and Implementation

A wearable WBAN system should be comfortable to wear. This indicates that the system should be as unobtrusive to the user as possible: it should be light, have a small footprint, and be mechanically flexible so it will not interfere with the user’s movements. The goal of this design is a system that meets the IEEE 802.11b/g/n standard requirements. Therefore, the button sensor antenna works at 2.45 GHz and 5.5 GHz WiFi bands. The reflection coefficient (S_11_) should be kept below −10 dB in the band. The radiation efficiency of a minimum of 65% was targeted. Many features, such as operating frequency band, radiation pattern, and impedance matching, should be steady and meet the requirements even when the antenna is placed on different body locations and sizes. To achieve maximum stability, it is necessary to reduce the interaction with the body. The reduction of the coupling to the body tissue not only keeps the antenna performance as if it works in free space, but also reduces the amount of energy absorbed by the wearer’s body and keeps the SAR levels down. In any practical scenario, for line-of-sight (LOS) and non-line-of-sight (NLOS), the BSA should be able to connect and interact with the system within a range of 40 m.

### 2.2. System Design Overview

The topology is shown in Figure 2. We used low-cost, flexible, and rigid dielectric substrates. The system consists of the following components: an ESP8266 EX Chip, a crystal oscillator, a Future Technology Devices International (FTDI) connector, a 4 MB flash memory, step-down regulators (3.3 V), a matching circuit, and a power supply. All the components are integrated on the PCB. A 50 Ω connector is used for the connection to the semi-flexible button antenna. All communications between the components occur via the serial peripheral interface (SPI) protocol and universal asynchronous receiver–transmitter (UART).

In the ESP8266 chip, the radio frequency (RF) transceiver supports the various channels of the IEEE802.11b/g/n standards. Furthermore, the impedance of the ESP8266EX power amplifier (PA) output is 39+j6 Ω (source impedance). Therefore, a matching circuit is added. The input impedance (load impedance) is 49−j4 Ω (from the button sensor antenna to the chip) after the addition of a T-shaped matching circuit consisting of two series inductors (1 nH) and a parallel capacitor (1 pF), connected to the ground plane. A U.FL connector connects the button antenna to the wireless sensor module via the matching network. In addition, a 1000 mAh Lipo rechargeable battery with a volume of 12 mm×4 mm×2 mm was chosen as the power source. A wrist band cover (similar to a watch cover) was manufactured by a 3D printer using polylactic acid (PLA) material. PLA is known to be harmless to humans and was used as the enclosure material for the sensor module. The BSA was programmed using the received signal strength indication (RSSI). The button sensor antenna was connected to the internet, to transmits the on-body monitoring data to the Blynk cloud in the Android using the TCP/IP protocols.

### 2.3. Button Antenna Topology and Approach

The process began with designing a button sensor antenna, as shown in Figure 3, for which simulations were conducted using CST Microwave Studio (MWS, Dassault Systems, Framingham, MA, USA). The antenna’s radiating part was constructed on the top PCB button Rogers substrate (RT/Duroid5880, Rogers Corporation, Chandler, AZ, USA). This PCB button substrate has a thickness of 1.574 mm, a relative permittivity of 2.2, and a loss tangent of 0.0009, which was placed on a 1.50 mm thick layer of felt substrate with a relative permittivity of 1.4 and a dielectric loss tangent of 0.044 S/m. There is a 3.76 mm air gap between the PCB button substrate and the felt layer substrate. The conductive ground layer is made of ShieldIt, with an area of 45 mm×45 mm, a thickness of 0.17 mm, and an estimated conductivity of $1.18\times 10^{5}\;S/\;m.$ The ShieldIt layer was glued on the bottom side of the felt substrate. The conductive parts on the top side of the PCB with a radius of 8 mm are as follows: a pin-fed patch on the bottom side of the PCB, and an asymmetrical capacitive patch connected to the bottom patch on the top side, which is also short-circuited to the ground plane by a shorting via. The coaxial feed and the shorting via have diameters of 1.27 mm and 1.22 mm, respectively. Manual cutting tools were used to cut the felt substrate and the ShieldIt ground plane. Glue was used to adhere the ShieldIt ground plane to the felt substrate. For the SMA connector and its galvanic connection to the ShieldIt layer, a through-hole was made. The conductive ShieldIt layer can resist temperatures of up to 250 °C; therefore, soldering was possible. In addition, the center asymmetrical slot was capacitively coupled to the patch, and the addition of an extra load staircase shape was utilized to achieve a compact size, which is discussed in [49]. The radiating patch located on the top side of the PCB can be considered as a circular loop. Therefore, its radius (a) can be calculated by:(1)$$a=\frac{F}{{\left\{1+\frac{2h}{\pi \varepsilon_{r}F}\left[\ln\left(\frac{\pi F}{2h}\right)+1.7726\right]\right\}}^{\frac{1}{2}}}$$
(2)$$F=\frac{8.791\times 10^{9}}{\sqrt[{f}]{\varepsilon_{r}}}$$
where h is the thickness of the substrate and $\varepsilon_{r}$ is its dielectric constant. Equation (1) does not take the fringing effect into account. Since fringing makes the patch electrically larger, the effective radius $\left(a_{e}\right)$ of the patch must be used, as given by [50].
(3)$$a_{e}=a{\left\{1+\frac{2h}{\pi \varepsilon_{r}a}\left[\ln\left(\frac{\pi a}{2h}\right)+1.7726\right]\right\}}^{1/2}$$
(4)$$f_{r}=\frac{1.8412v_{0}}{2\pi a_{e}\sqrt{\varepsilon_{r}}}$$
where $v_{0}$ is the free-space speed of light and $f_{r}$ is the resonance frequency.

### 2.4. Design of Printed Circuit Board

The Eagle software version 7.7.0 design tool was used to create the schematic design and a printed circuit board. After validating the simulation and breadboard prototyping, the schematic design was transferred to a PCB prototype, as shown in Figure 4. To keep the schematic design as small as possible, surface mount components are used. The diameter of this PCB is 45 mm. The prototype was optimized and designed on FR4 material, which has a dielectric constant of 4.2, a loss tangent of 0.02 S/m, and a thickness of 1.6 mm. The developed wireless sensor module was then properly programmed for RSSI at the ISM band at 2.45 GHz using the IEEE 802 b/g/n standard and incorporated with a BSA via a T-shaped matching circuit, as shown in Figure 5. ESP8266 is a WiFi-enabled microcontroller integrated circuit. The wireless modules are widely utilized to control devices via the internet. The ESP8266 can be programmed using the Arduino IDE3 in C++, the firmware Espruino4 in JavaScript, and the Espressif SDK in C. For Internet of Things (IoT) applications, ESP8266 has become a popular and affordable solution. The 26 MHz Crystal Oscillator, 4 MB Flash memory, USB to FTDI connector, stepdown regulator (3.3 V), power supply, and 2.4 GHz RF transceiver are all included in the ESP8266 EX wireless module (Espressif Systems, Shanghai, China).

### 2.5. Impedance Matching

The input impedance is a fundamental parameter that influences the reflection coefficient (S_11_) as well as the impedance bandwidth (BW). This section describes the design of a matching network that will decrease the reflection coefficient (S_11_) level at the desired resonance frequency while increasing the impedance bandwidth. The matching circuit ensures maximum power transmission from the source (A) to the button antenna (C) and maintains the required performance even when detuning occurs (Figure 5).

Input impedance ($Z_{in}$) was measured using a vector network analyzer (VNA), which was then utilized to create an optimum matching network. A T-shaped matching network was used, as shown in Figure 5. CST MWS was used to optimize the T-shaped matching network. By modifying the values of circuit components C_1_, L_1_, and L_2_, the matching network optimization was performed to reduce the power reflection at the button antenna’s input port. Table 2 shows the final realized values. At the button antenna input, a T-shaped matching circuit with two inductors (1 nH) and a capacitor (1 pF) cancels the reactive part of the impedance and gives an excellent match to 50 Ω with an impedance bandwidth ($S_{11}<-10\;\mathrm{dB}$) in both frequency bands. Furthermore, the antenna produced an omnidirectional 3D radiation pattern at 2.45 GHz.

### 2.6. Enclosure Design

We chose an enclosure similar to a hand-watch case. Figure 6 shows the hand-watch case designed in the 3D Tinkercad simulation tool. The 3D model has an area of 46 mm×46 mm and is divided into two parts: the upper part of the volume is used for the main PCB; the lower part is used for a rechargeable Lipo battery. A power switch hole, a USB porthole, and a sensor hole were incorporated into the upper space. The 3D enclosure was fabricated using a 3D printer and PLA material. PLA is biocompatible and safe for humans. The button antenna was placed on the top part of the enclosure, and they were connected through a U.FL connector.

## 3. Results and Discussion

To find the optimum design parameters, a parametric study was conducted while yielding a realizable structure. The time–domain solver in CST MWS was used for the design and parametric tolerance analysis. Simulations and measurements were performed in the frequency band of 1–7 GHz.

### 3.1. Evaluation of Button Sensor Antenna: Simulated Results

Several measurements and simulations were conducted to examine the performance of the wearable button antenna. These included examinations for robustness, durability, bending, and crumpling impact. A layered cubic phantom containing skin, fat, muscle, and bone with the proper electrical properties of human tissues was created. Similarly, a cylindrical arm phantom with a radius of (R) is introduced for bending investigations. These models were made in CST MWS to study the BSA performance in on-body conditions. Our three-layer standard body model has a volume of $200\times 200\times 50\;{\mathrm{mm}}^{3}$ (skin = 4, fat = 8, muscle = 30, and air layer under the tissue layers = 8 (in mm)) and the four-layer phantom model has the volume of $200\times 200\times 88\;{\mathrm{mm}}^{3}$ (skin = 2, fat = 8, muscle = 40, bone = 30, and air layer under the tissue layers = 8 (in mm)). The performance of the designed BSA device was evaluated for different tissue layer morphologies as well as spacing from the phantom model, according to the standard methods discussed in [51,52,53,54,55].

#### 3.1.1. Button Sensor Antenna Feed Locations

Four feeding positions on the chassis were examined: the center, the long edge, the short edge, and the corner. A ground plane (45 mm×45 mm) is used. The lower frequency range is served by the imaginary part of the input impedance (X_in_) in a button sensor antenna. When the resistance (R_in_) is low, it signifies that the design is difficult to match to a 50 Ω input impedance and, as a result, the antenna cannot efficiently radiate. When the button sensor antenna’s feed point is placed at the center, short edge, or corner of the ground plane, its lower (upper) resonance frequency moves to 2.61 (6.20) GHz, 2.42 (6.00) GHz, 2.31 (7.00) GHz, respectively. In all cases, the BSA is not matched at 5.6 GHz. The resistive part of the input impedance is 26.92 (5.00) Ω, 42.32 (13.00) Ω, and 11.00 (29.00) Ω, respectively. The button sensor antenna’s resonance is moved to 2.45 GHz when the feed point is located at the long edge. In this location, the input resistance is 51.12 (47.42) Ω, at the 2.45 GHz lower and 5.6 GHz higher bands, respectively. The long-edge location provided S_11_ bandwidth (BW) lower than −10 dB of 2.42 GHz to 2.47 GHz and 5.37 GHz; these were called the lower and upper bands. The coupling between the top button and the textile ground causes changes in input impedance for different positions. The long-edge location produces a strong ground coupling. In this case, the ground plane works like a radiator, increasing the resistive part of the input impedance. Furthermore, the simulation shows that the x- (horizontal) and y- (vertical) components of currents are out of phase at the center, short edge, and corner. These components cancel each other and do not contribute much to the improvement of the antenna’s impedance bandwidth. On the other hand, the x- (horizontal) and y- (vertical) components of the surface current are in phase in the long-edge corner, creating a strong radiated field.

Figure 7a depicts the surface current at the lower band that follows the A-B-C path ($l_{1}$). It shows that the current path is half a wavelength at 2.4 GHz, resulting in an omnidirectional radiation pattern. Figure 7b depicts the current route at the higher band that follows the A-B-D pattern ($l_{2}$). This current takes a 1/4 wavelength path at 5.6 GHz. It also shows a significant amount of current on the ground plane, which will perform as a radiator due to shorting pin. Therefore, the modes depend on the top patch, bottom patch, and shorting pin. Figure 8 depicts the simulated reflection coefficients (S_11_) of the button sensor antenna (BSA), as well as the impact of feeding locations on the S_11_. It can be seen how changing the button sensor antenna’s feeding point has a considerable impact on the impedance matching level and resonant frequency bands. Due to the presence of the lossy tissues, the radiation pattern shows a wider beam with a wider bandwidth for the on-body case. As illustrated in Figure 9, omnidirectional radiation patterns are obtained.

#### 3.1.2. Human Body Phantoms

A button sensor antenna design is wearable; however, it is on a solid platform that provides better rigidity and stability than a textile. In healthcare applications, this capability is advantageous. Figure 10 depicts various body phantoms and antenna locations that were used in the simulations to characterize the performance of the wearable antenna.

#### 3.1.3. Three-Layer and Single-Layer Body Phantom Models

To deploy wearable designs for body monitoring applications, it is first necessary to study their performance while being placed on the body and the lossy tissues. Some of the models of interest are the chest and the arm. In this part, the BSA performance on the human chest and arm models will be discussed. A flat 3D-layered body model with skin, fat, and muscle layers was mostly used for the chest model as well thickness variation in biological tissues, as discussed in [54]. The overall volume of the three-layer phantom is $200\times 200\times 50\;{\mathrm{mm}}^{3}$ and the thicknesses of the layers are as follows (in mm): skin: 4, fat: 8, muscle: 30, and air layer under the tissue layers: 8, for a total thickness of 50 mm. To consider the effect of bone, we also considered a four-layer phantom which has slightly different layers and thicknesses. After a layer of bone is added, the overall volume of the phantom is $200\times 200\times 88\;{\mathrm{mm}}^{3}$. The layers thicknesses are as follows in (mm): skin: 2, fat: 8, muscle: 40, bone: 30, and air layer under the tissue layers: 8, for a total thickness of 88 mm. The skin has a dielectric constant (ε_r_) of 31.29 and conductivity (σ) of 5.0 S/m at 2.45 GHz, for the fat layer the values are 5.28 and 0.1 S/m; whereas at 5.8 GHz, the values for skin are 35 and 3.7 S/m and the values for are 5.0 and 0.3 S/m, respectively. The muscle layer has an ε_r_ of 52.7 and 48.2 and a σ of 1.95 S/m and 6.0 S/m at the lower and upper-frequency bands, respectively. The bone layer has an ε_r_ of and a σ 11.4 of 0.39 S/m, and of 9.674 and 1.1544 S/m at 2.45 and 5.8 GHz in the lower and upper frequencies, respectively [55,56,57,58]. For modeling the arm, a simple layered cylindrical model was utilized. For the on-body arm communication, a phantom with a radius (r) of 50 mm and 100 mm and a length of 150 mm was chosen. Figure 10 shows these models and the results of the simulations. Generally, the analysis of the performance of a wearable antenna is difficult because the dielectric properties of the human tissues are dispersive and change with frequency. In addition, the tissue properties from one person to another may be different. To add to this difficulty, the body size is different for each person. In the three-layer model, we varied the layer thicknesses [59,60].

Another important parameter to consider is the air gap between the skin surface and the bottom of the button antenna surface. The variation in the air gap may have a significant impact on wearable device functionality. We evaluated an air gap of 2 mm, 3 mm, and 5 mm. The resonant frequency and impedance bandwidth in both bands were affected by the air gap size. We present this effect through different simulations. Although we used an air gap with varying size, this may also mimic the combination of several textile layers and air, since textile and air have similar dielectric constants. Because of the significant loading effect of the body tissues on the BSA at the air gap of d=2 mm, the impedance matching was affected. The impedance matching was unaffected by increasing the air gap to 5 mm, and the BSA was almost decoupled from the body at this distance. We recommended a 5 mm air gap and kept this value for the rest of the studies. S_11_ at different air gaps is shown in Figure 11. A similar simulation was set up for a single-layer muscle phantom. Figure 12 depicts the single-layer model of a 50 mm and 88 mm single layer of muscle, and air layer under the tissue layers, with the total volume of $200\times 200\times 50{\;\mathrm{mm}}^{3}$, and $200\times 200\times 88{\;\mathrm{mm}}^{3}$. Similarly, for the arm model, a cylinder of homogenous muscle phantom with a radius of 50 m and 100 mm and a length of 150 mm was considered. We again observe that with a minimum air gap of 5 mm between the body and the BSA design, the effect of body tissues is minimum, and the performance of the BSA becomes similar to that of free space, as indicated in [52,56,61,62]. The results reveal that the architecture behaves similarly in the presence of a variety of phantom body models. These results attest to the suggested design’s robustness. By keeping the air gap at 5 mm or more, the design becomes less affected by the variation in the tissue properties.

#### 3.1.4. Effects of Fabric

A layer of fabric might normally be under or over the BSA. In order to take the effect of fabric into account, various garments were considered, and the antenna was simulated while the fabric layer was underneath and over the antenna and covering it. These scenarios are illustrated in Figure 13. The distance between the surface of the antenna and the fabric was chosen to be 1 mm. A fabric thickness of 5 mm was considered. Various typical materials were considered in the simulations [63]. The reflection coefficient (S_11_) for the cases where the fabric was on top of the BSA is shown in Figure 14. Figure 15 depicts the reflection coefficient (S_11_) for the cases that the fabric was placed under the BSA. In both cases, the system continues to function effectively and efficiently within the specified operating frequency bands, demonstrating the proposed design’s robustness.

#### 3.1.5. Bending Effects

The various possible bending conditions that we considered are depicted in Figure 16. When the textile substrate is bent or comes into contact with body tissue, it must maintain its resilience. Depending on the fabric’s thickness, the resonance frequency may be moved due to the varying distance from the body tissue layers. Three bending fixtures were employed in this part to study: (a) free space, (b) on the flat phantom model, and (c) on the cylindrical phantom model. The results are compared with those from the flat condition. Figure 17 shows the comparisons of S_11_ at various bending radii. The lower band (half wavelength) shifted from 2.45 to 2.345 GHz, whereas the higher band (quarter wavelength) does not show much variation in the frequency band. The frequency band is from 2.350 GHz to 2.45 GHz and 5.0 GHz to 6.0 GHz when the bending diameter (D) is 50 mm, resulting in small changes in the resonance frequency.

As illustrated in Figure 18, the radiation patterns in both bands are investigated under various bending conditions. The results revealed that the radiation patterns are practically impervious to structural deformation. The two cases of the design bent on the chest and arm phantoms were investigated. The shift in resonance frequencies and bandwidths is smaller in the case of bending in free space than when the antenna is bent on the chest and arm body phantoms with different sizes. Nonetheless, the BSA operates in the appropriate frequency bands. Therefore, we can conclude that the BSA has robustness against a bending radius of R=50 mm. Please note that we did not include any stretching in the simulations.

For a substrate with a thickness (h) of approximately 1 mm, the compression of the substrate is around 1% for a bending radius (R) of 35 mm. We consider the bending for a thin substrate; therefore, neglecting the substrate compression is an acceptable approximation [64]. This topology ensures robustness and minimum effects due to bending, stretching, and wet situation. The proposed BSA design was simulated in this research, and performance tests on body bending models were conducted to confirm the design system performance. Therefore, the design performance cannot be affected by the human body, nor can the human body be affected by the design. The antenna’s performance is minimally affected by bending, movement, and fabrication tolerances. Therefore, the proposed design shows robustness.

### 3.2. Measurement Results

The BSA was tested in free space and on the human body. A network analyzer (Keysight PNA-X N5242A, Keysight, Santa Rosa, CA, USA) was used to measure S_11_ after calibration by a Keysight calibration module (NA4691B, Keysight, Santa Rosa, CA, USA). A flat human homogeneous tissue phantom with dimensions of $200\times 200\times 50\;{\mathrm{mm}}^{3}$ was utilized to assess the performance. An air gap similar to the one assumed in the simulations (5 mm) was maintained between the surface of the phantom and the antenna ground. This air gap is in a practical range and is also required for safety reasons. The constructed device was also tested on the human subjects’ bodies. The body affects the resonance frequency and radiation properties, but the operating frequency bands were still maintained even in the presence of the human body.

#### 3.2.1. Reflection Coefficient and Impedance Bandwidth

Figure 19 depicts the measured S_11_ for the BSA in free space, including on the human chest and arm phantoms. Simulated and measurement results are in good agreement. The BSA was tested on two human bodies with two diverse weights of up to 65 and 110 (in kg), and it performed similarly to that measured in the free space. Initially, the BSA was almost flat when placed on the chest. The flatness of the body, the air gap between the body and the BSA, and any layers between the body and the BSA created a similar condition to the free-space scenario. The air gap was kept around 5 mm. Similarly, for the second round of testing, the BSA was wrapped around the arm while retaining the same air gap as before. The measured S parameter showed that the impedance bandwidth was not significantly altered. Bending of the BSA and body humidity (sweating) are two other situations that could cause changes in performance. In these two conditions, the lower and upper frequency bands were still covered by the BSA, and no significant variations in the reflection coefficient (S_11_) were observed.

The investigation was conducted at several locations under bending conditions. The BSA was measured in free space as well as on polystyrene cylinders with a dielectric constant of ε_r_ ≈ 1 and diameters of D (in mm)=100, 80, 70, 60, and 50. The different diameters produce different curvatures, which were used to see if the required frequency range is maintained whilst bending. Therefore, there was a slight shift in the resonance frequency. As illustrated in Figure 20, the measured S_11_ shows that the antenna is matched well within the desired frequency bands.

The humidity of the body may have a big impact on wearable antennas. As indicated in Figure 21, the proposed work investigated the case when the BSA was dampened by moisture by applying water on the textile layer until it was fully saturated. The damp condition had a significant impact on S_11_. Water evaporation from the entire substrate may take up to 3 h. The results show that after 1 h and 2 h, the center frequency is still shifted and that after 3 h, the BSA returned to dry conditions with a room temperature of 26 °C and a 30% air humidity.

#### 3.2.2. On-Body Communication System

The BSA’s transmission characteristics for on-body monitoring activity were tested in two locations: the chest and the arm, in both line-of-sight (LOS) and non-line-of-sight (NLOS) scenarios.

#### Measurement Setup

The measurement set up for the path loss is shown in Figure 22. The BSA was used as a receiver antenna and was mounted on the body phantom on a wooden tripod at a distance of at least 1.5 m (m) from the ground and 1.7 m from any wall. The BSA link budget was assessed in free space as well as in on-body models in both indoor and outdoor environments. The platform was rotated to perform measurements in various orientations. The measurement was performed at 2.45 GHz to evaluate the path loss over various distances. For the LOS and NLOS situations, measurements were taken at various distances (d) and positions between the gateway (A) and the receiving BSA (B). In both cases, the measuring environment was roughly 20 m wide. The transmission coefficient S_21_ was measured via the LED/receiving signal strength programming code, which was measured between the BSA and the WiFi router. The spectrum analyzer (MS2720T) with zero spans was also connected to the BSA. The gateway (A) was mounted to the wall, while the BSA (B) was moved up to 40 m to assess the path loss (S_21_). It was determined that all the surrounding objects were in a stationary state and that the measurement was performed correctly. To clear the ripple effects due to the cables and the measurement equipment, a two-port calibration was performed. For on-body link budget calculation, we used a Blynk platform, using iOS and Android apps, to control BSA devices over the internet. We installed the Android Blynk App to display the output data of the BSA device at 2.4 GHz because WiFi routers operate at 2.4 to 2.48 GHz. The signal strength was tested via the LED/receiving signal strength using programming codes in the wireless sensor module to track the body location with a maximum range of 40 m (see Appendix B). Therefore, we uploaded the codes for LED/RSSI at the GitHub repository [65].

#### 3.2.3. Radiation Patterns Measurement

As illustrated in Figure 23, the far fields were tested inside the anechoic chamber. The BSA was mounted on a body phantom that served as a receiving device, while the horn-shaped antenna served as a transmitting antenna, moving in the azimuth and elevation planes. Keysight vector network analyzer model PNA-X N5242A was used to measure S_21_ and the radiation patterns. The dielectric properties of the phantom are similar to human muscle tissue, with a relative permittivity (ε_r_) of 52.7 and 48.2, and a conductivity of 1.95 and 6 S/m, at the lower and higher bands, respectively [66]. Figure 24 shows the predicted measured radiation patterns for different cases and frequencies. The arm is usually a good place for mounting wearable antennas because the radiation in horizontal positions from the arm is less influenced than the radiation from the antennas placed on the chest. The first location is less surrounded by the tissues than the second. At the lower band, it can be seen that the BSA has an omnidirectional radiation pattern, which can be employed for on-body communication. At the upper band, omnidirectional radiation was also achieved, which could be useful for off-body communication. Fabrication tolerances and imperfections of the materials might impair the performance of textile antennas in real-world circumstances. As can be seen, the BSA maintained a good radiation pattern stability under different conditions at different frequency bands.

#### 3.2.4. On Body Link Budget Calculation

It is very important to examine the link budget of the wearable devices in body-centric communications to provide a stable link budget, as shown in Figure 25. We tested S_21_ in free-space and on-body communication cases in an open environment (or residential area). This test allowed the research work to demonstrate the range of expected values around the path loss. The path loss (PL) is defined as the ratio of the input power (at point 1) to the power received (at point 2) that is measured in terms of the transmission coefficient (|S_21_|)).
(5)$$PL_{dB}\left(d\right)=-{\left|S_{21}\right|}_{dB}$$

#### Free-Space Communication

Path loss measurements were first performed in free space as a baseline, as illustrated in Figure 26. The BSA was placed on a tripod at the height of 1 m above the ground, with the WiFi router vertically aligned. Furthermore, the distance (d) was chosen at 2 m to 40 m in the S_21_ measurement to estimate the path loss in LOS and NLOS. Please note that a full-wave simulation was not possible; therefore, we used the open space area in an inside and outside-the-house measurement to precisely calculate the path loss exponent. The log–distance path-loss model was used, which can be written as:(6)$$\;PL_{dB}\left(d\right)=PL\left(d_{0}\right)+10nlog\left(\frac{d}{d_{0}}\right)$$
where n is the path loss exponent, $d_{0}$ is the close-in reference distance and d is the far point representing the separation between transmitter (Tx) and receiver (Rx) [67]. After substituting the values $d_{0}=2\;m$ and d=40 m, we obtained an average range for n∈[2.7 to 3.5] for a distance of 10 m to 11 m, and a range of n∈ [4.0 to 6.0] for the distances of 11 m to 40 m, which were well matched with the urban areas and obstructed building paths, respectively [67].

#### On-Body Communication

In the cases of an on-body situation, measurements were taken on a 2.45 band. As shown in Figure 27, the BSA was located on body locations (chest and arm) to calculate S_21_. To explore the influence of distance (d) on S_21_, the separation between the BSA and WiFi router was adjusted to 2, 5, 8, and 40 (in m), with a 5 mm gap was created between the BSA and the top skin layer. The BSA was located on the chest of male volunteers with a weight of 72 kg and a height of 1.58 m, both in LOS and NLOS, inside and outside of the building. In the first sets of measurements, the subject’s arms were spread across the subject’s body. However, due to the power absorption of the body tissues, the path losses were higher in the presence of the body, in LOS and NLOS, compared to free-space measurements. For the on-body chest communications, the distance (d) between the design and WiFi router was raised from 2 m to 40 m, similar to the free-space measurements. We computed the path loss values (n), which are slightly greater than the free-space case, using (8) and $d_{0}=2\;m$ and d=40 m. Because of the restriction caused by the bodies, as well as the losses and power absorbed by the body tissues, the increase in losses were expected. The BSA was then mounted to the male subjects’ arms in LOS and NLOS, and path loss was measured. The path loss was found to be 72 dB. The BSA was tested in the NLoS in the outdoor environment in the following situations, where it was put on-body chest and the arm of a male subject, at the distance of 40 m, 84 dB path loss was measured. Therefore, the path losses in numerous situations of free space, on-chest, and on-arm, both in LOS and NLOS in indoor and outdoor environments, were not more than 84 dB within the maximum range of 40 m. To comply with the SAR regulations [68], the input power was adjusted to a maximum of 100 dBm (0.5 W) (SAR values are discussed in the next section). When the receiver demonstrated a received power of −75 dBm [69], then the maximum loss of 99.4 dB is acceptable. Thus, the proposed design offers good performance and can be used for LOS and NLOS situations in indoor and outdoor environments.

#### 3.2.5. Specific Absorption Rate

SAR measurement is a necessary element for health safety in WBAN systems. The SAR depicts how electromagnetic radiation is absorbed by tissues, causing a temperature increase. As a result, the SAR levels are limited to protect safety the human body from thermal radiation. Therefore, SAR is linked to E-fields and tissue conductivity [43]:(7)$$\mathrm{SAR}=\frac{\sigma {\left|E\right|}^{2}}{\rho }\left(\frac{W}{\mathrm{Kg}}\right)\;$$
where σ shows the tissue conductivity and ρ is its density. Alternatively, SAR shows the time derivative of the dissipated absorbed energy (dW) in mass (dm) in a volume (dV) of a provided density (ρ).
(8)$$\mathrm{SAR}=\frac{d}{\mathrm{dt}}\left(\frac{\mathrm{dW}}{\mathrm{dm}}\right)⟹\frac{d}{\mathrm{dt}}\left(\frac{\mathrm{dW}}{\rho \left(\mathrm{dV}\right)}\right)\;$$
(9)$$\mathrm{SAR}=\frac{1}{\rho }\left(\frac{\mathrm{dW}}{\mathrm{dt}}\right)\;.\;$$

The SAR values are calculated for an average of 10 g of tissues and should be less than 2 W/Kg based on the “International Commission on Non-Ionizing Radiation Protection” (ICNIRP, European) standard. The SAR values will be higher if the BSA is located near the body’s skin. In measurement, the gap between the body and the bottom of the BSA was at 5 mm to be comparable to the simulation. The air gap and the ground plane reduced the SAR. Similarly, for a 1 g tissue average, the “Federal Communication Commission” standard (FCC, and US standard) limit is 1.6 W/kg. The SAR values in 1 g and 10 g tissue average for lower and upper bands, were calculated, and they were lower than the standard at an input power of up to 0.5 W (100 dBm). Maps of 1 g and 10 g average SAR values for various situations at the maximum input power of 0.1 W or 0.5 W for the on-body chest and arm cases in both bands are demonstrated in Figure 28. Therefore, simulated verification of the SAR has been examined for flat and bending cases, and similar results have been obtained. Finally, although the study is based on the device prototypes, the results are generally valid for various wearable devices. Table 3 provides the simulated SAR values (using CST MWS) for the flat situation. Whereas Table 4 shows the bending cases. Finally, the characteristic features of BSA for flat and bending situations have been briefly summarized in Table 5 and Table 6, respectively.

## 4. Conclusions

In this work, a button sensor antenna was presented along with a wearable wireless sensor module for WBAN applications working at two frequency bands, of 2.45 GHz and 5.6 GHz. The device is small, button-shaped, with an area of 45 × 45 mm^2^, and it can be fitted into a 45 mm diameter circular-shaped wireless sensor module that is enclosed into a 3D enclosure as large as a hand-watch cover for users’ convenience. Combining the rigid button PCB that included the wireless sensor module with the textile layers in the proposed design provides several benefits, including high stability in operation resonance frequency, impedance matching level, radiation patterns, less path loss, less attenuation through materials, improved wireless communication range, and reduced SAR values. The simulation and measurement results validate the concept of making a button antenna that provides wireless connectivity in a relatively small package. The unique aspect of this work is that it provides a systematic approach to the design of a complete transceiver in a wearable package with the option of wearing it as a button or a wristband, and the small package includes everything needed for communication with WiFi routers. While the wearable antenna is the heart of this design, the transmit/receive module with the matching elements completes it as a standalone unit. The unit is compatible with IEEE 802.11 b/g/n standards. The sensor is characterized for various conditions and is shown to be an effective and good candidate for WBAN health monitoring systems.

For the on-body (chest and arm) positions, we measured the reflection coefficient (S_11_) of −23.07 (−27.07) dB and −30.76 (−31.12) dB, and gain of 2.09 (6.7) dBi, and 2.16 (5.67) dBi, and radiation efficiency of 65.12 (81.63)% and 75.0 (85.0)%, for the chest and arm locations, in lower (upper) bands, respectively. In addition, the performance of the button sensor antenna was tested under extreme bending and wet conditions, and the design was found to work within the specified operating band. The 10 g and 1 g average tissue SAR tests were performed, and the results were within the standard limits. The measured results showed that the 2.45 GHz device platform has a communication range of 40 m, which is better than many other wearable devices used in healthcare reported in the literature. Through simulations and measurements, we have shown that the proposed design platform has the potential to be used in future wearable health monitoring applications. An example of these applications is a novel body-worn hearing aid for the elderly that is based on WBAN systems and utilizes beamforming. It is a hearing aid featuring 5G beamforming hearing technology, which helps in speech recognition in noisy environments, with advantages of highly concentrated coverage, energy-saving technology, and long-distance transmission [70,71].

## Figures and Tables

**Figure 1 micromachines-13-00475-f001:**
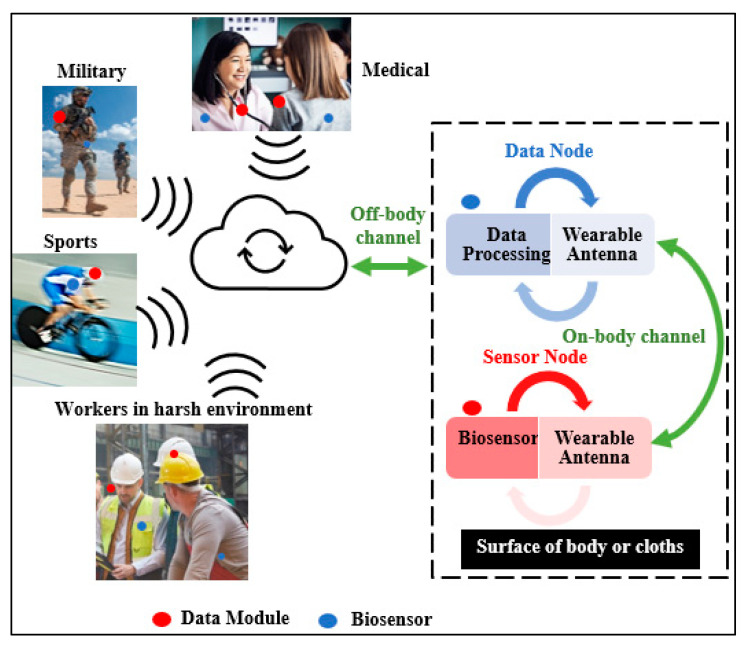
The general architecture of WBAN in healthcare applications.

**Figure 2 micromachines-13-00475-f002:**
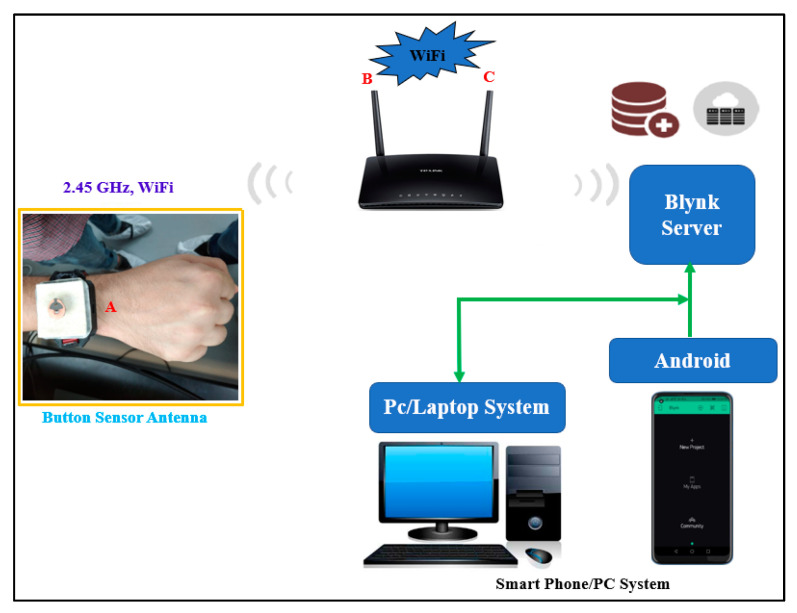
Design system architecture for the button sensor antenna (Point A: BSA Device, and Points B, C: WiFi Router).

**Figure 3 micromachines-13-00475-f003:**
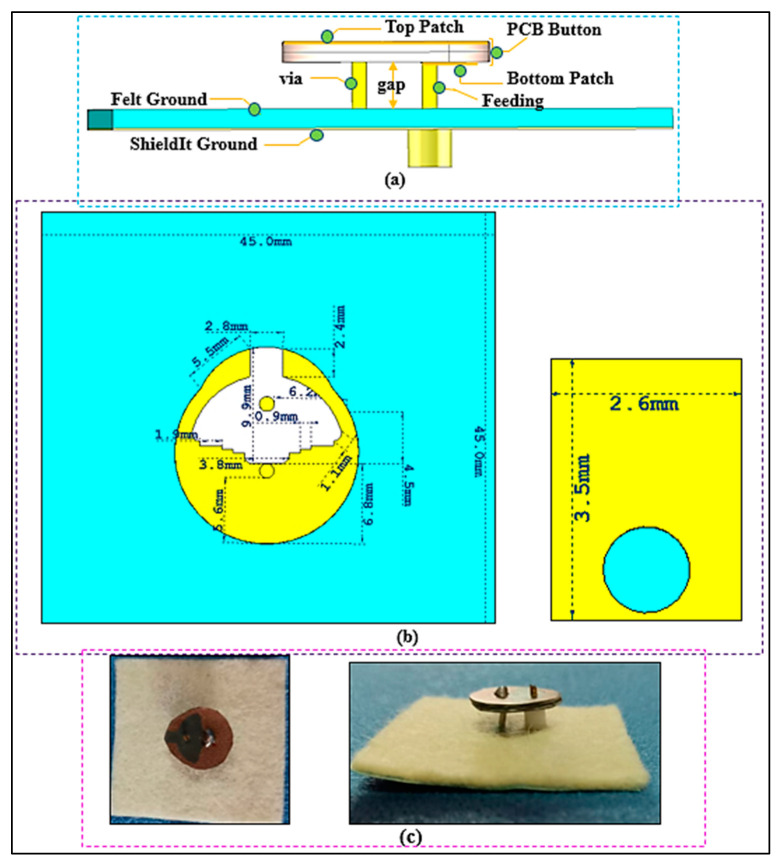
Structure and dimensions of the BSA (dimensions in the given mm): (**a**) the side view; (**b**) the top view; and (**c**) the fabricated BSA.

**Figure 4 micromachines-13-00475-f004:**
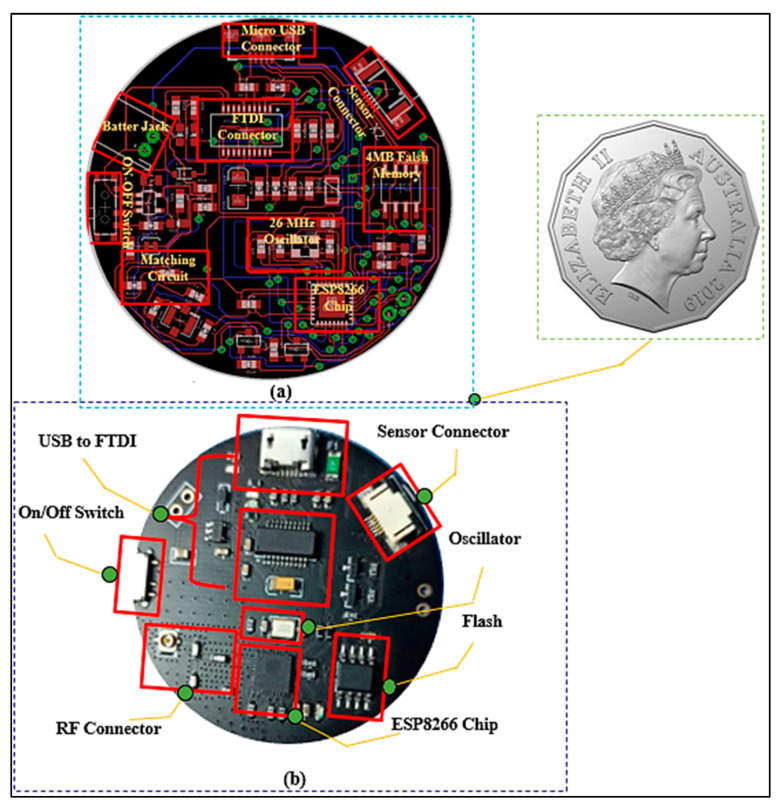
PCB Design. (**a**) PCB layout, and (**b**) fabricated prototype.

**Figure 5 micromachines-13-00475-f005:**
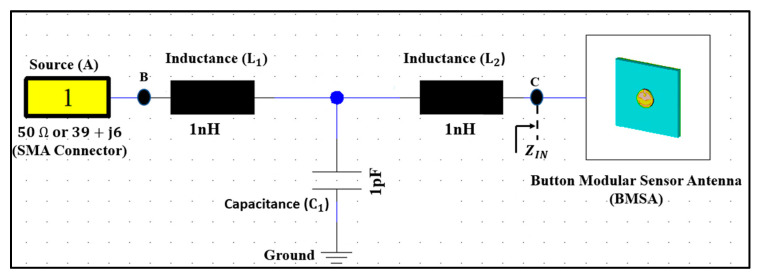
T-matching circuit (Z_IN_: 50 Ω).

**Figure 6 micromachines-13-00475-f006:**
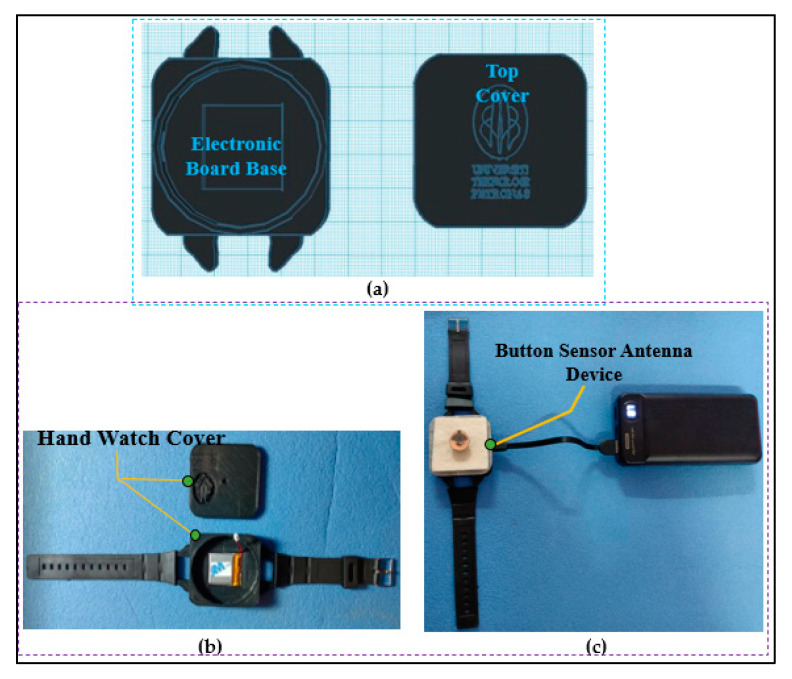
3D enclosure of a wearable button sensor antenna. (**a**) CAD design, (**b**) fabricated enclosure, and (**c**) full fabricated BSA prototype.

**Figure 7 micromachines-13-00475-f007:**
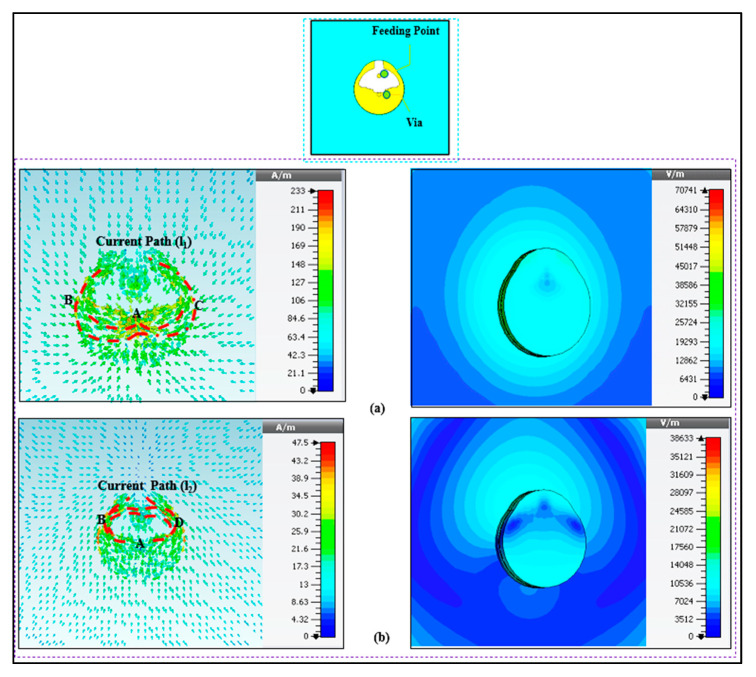
(**a**) Surface current distribution (left side) and magnitude (right side) at lower band 2.45; (**b**) Surface current distribution (left side), and magnitude (right side) at the upper bands at 5.6 GHz.

**Figure 8 micromachines-13-00475-f008:**
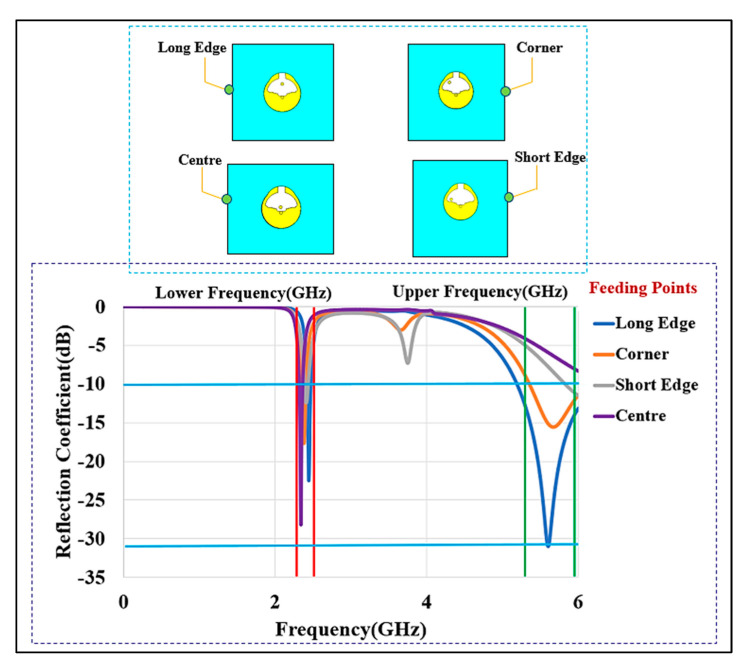
The reflection coefficient (S_11_) of the BSA for different feed locations (lower band markers are at 2.22 to 2.8 GHz; upper band markers are at 5.0 to 5.9 GHz).

**Figure 9 micromachines-13-00475-f009:**
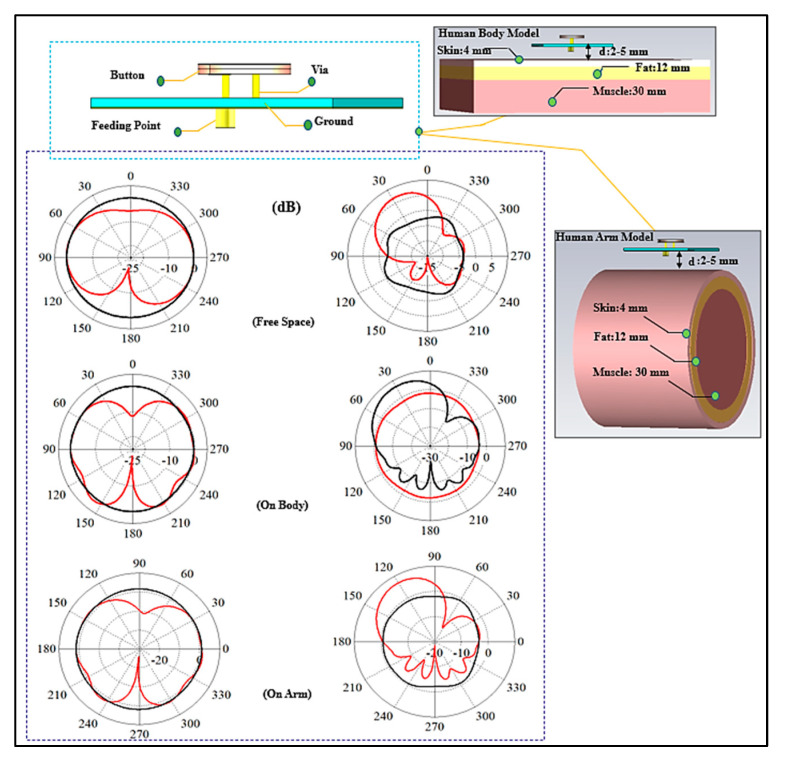
Radiation patterns of BSA on 3-layer phantom models at lower (**left**) and upper (**right**) bands in various scenarios (Solid black line, H plane; red line, E plane, at a gap of 5 mm).

**Figure 10 micromachines-13-00475-f010:**
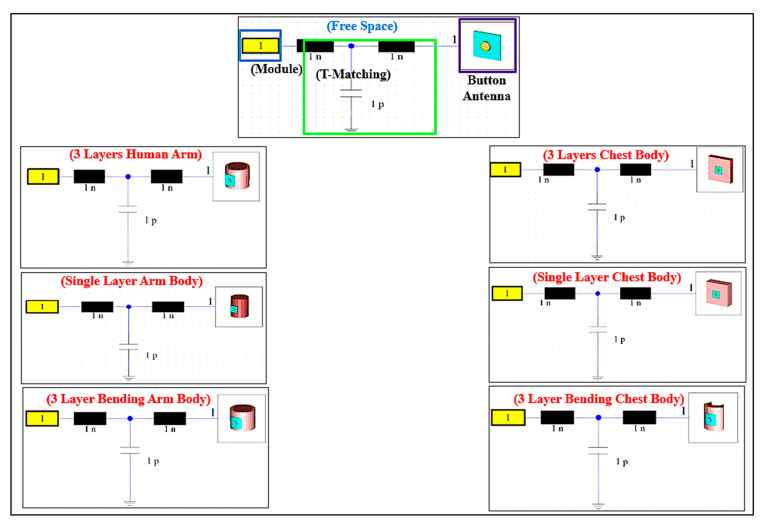
Set up for the 3-layer on-body cases considered for various testing purposes.

**Figure 11 micromachines-13-00475-f011:**
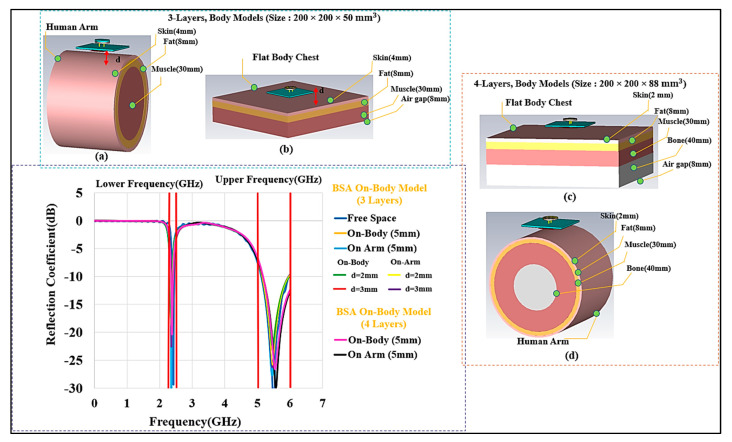
S_11_ for BSA. (**b**,**c**) on the body chest (flat), with a thickness of 50 mm and 88 mm, respectively, and (**a**,**d**) arm cylindrical models, with a radius of 50 mm and 100 mm.

**Figure 12 micromachines-13-00475-f012:**
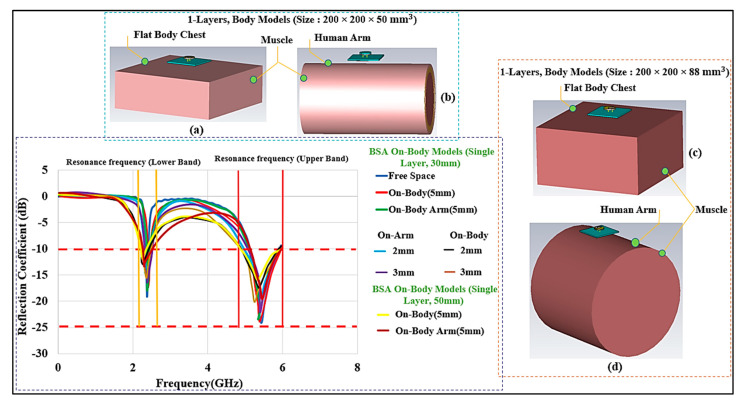
S_11_ for BSA. (**a**,**c**) Homogeneous model ($\mathrm{Muslce}\;\left(\varepsilon_{r}\right)$ = 56.6 and *σ* = 1.33 S/m, thickness 50 and 88 mm), and (**b**,**d**) simplified arm models for radius (r) of 50 mm and 100 mm for the small and large size bodies, respectively.

**Figure 13 micromachines-13-00475-f013:**
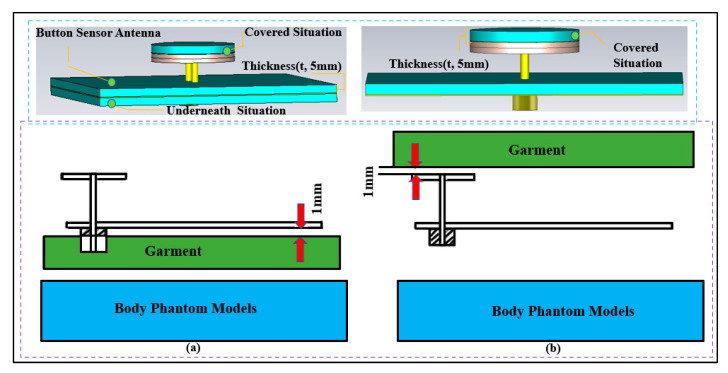
Button sensor antenna with the fabric placed (**a**) under the antenna, and (**b**) over the antenna (fabric thickness t: 5 mm).

**Figure 14 micromachines-13-00475-f014:**
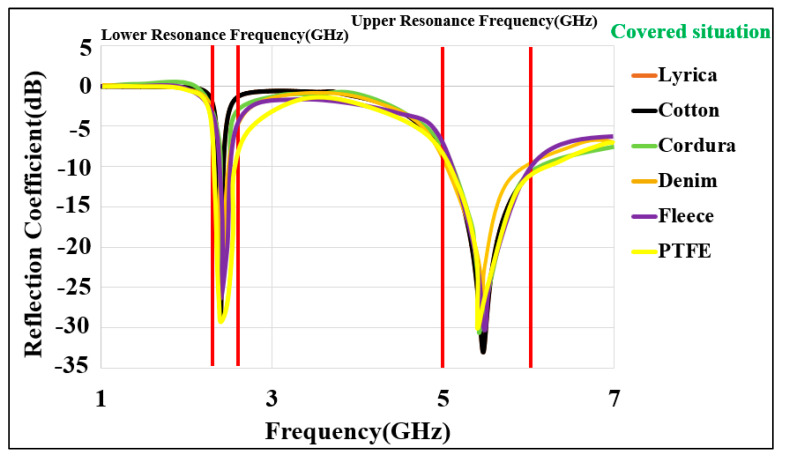
Reflection coefficient for the antenna covered by various types of fabric. (Textile fabrics relative permittivity (ε_r_): Cordura/Lycra: 1.50, denim: 1.47, fleece: 1.02, cotton: 1.60, and PTFE: 2.1).

**Figure 15 micromachines-13-00475-f015:**
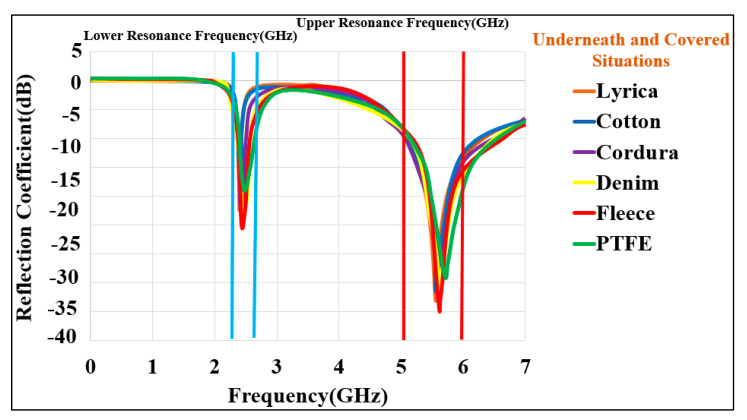
Reflection coefficient (S_11_) for the proposed BSA design covering situations using various types of flexible and fabric substrate materials.

**Figure 16 micromachines-13-00475-f016:**
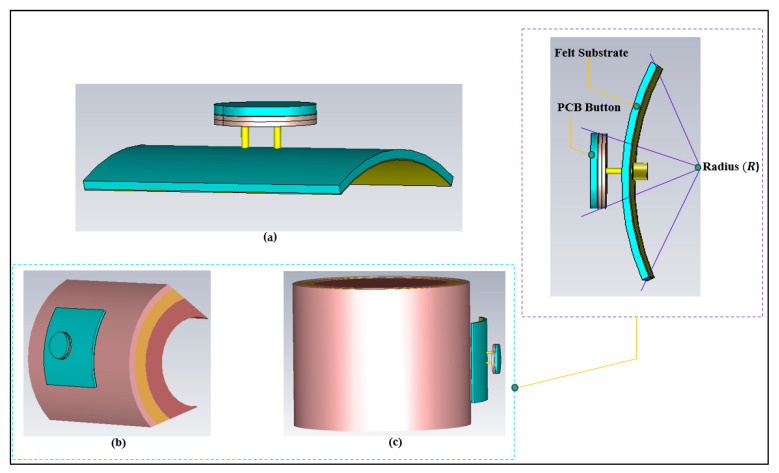
Study cases of the effect of bending of the BSA. (**a**) free space situation, (**b**) flat layered chest phantom (standard size, $200\times 200\times 50{\;\mathrm{mm}}^{3}$), and (**c**) arm cylindrical phantom (50 mm radius and 150 mm length).

**Figure 17 micromachines-13-00475-f017:**
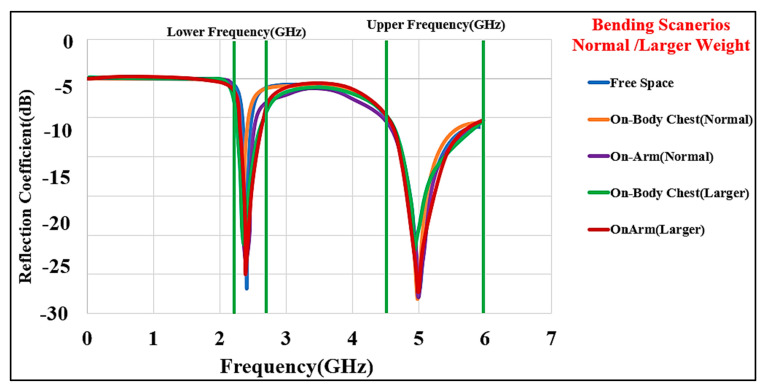
Simulated S_11_ for different bending cases. Chest, normal size $100\times 100\times 50\;{\mathrm{mm}}^{3}$, and large size $200\times 200\times 50\;{\mathrm{mm}}^{3}$; arm, small size 50 mm diameter, length (*l*), 100 mm, and large size 100 mm diameter, length (*l*), 150 mm.

**Figure 18 micromachines-13-00475-f018:**
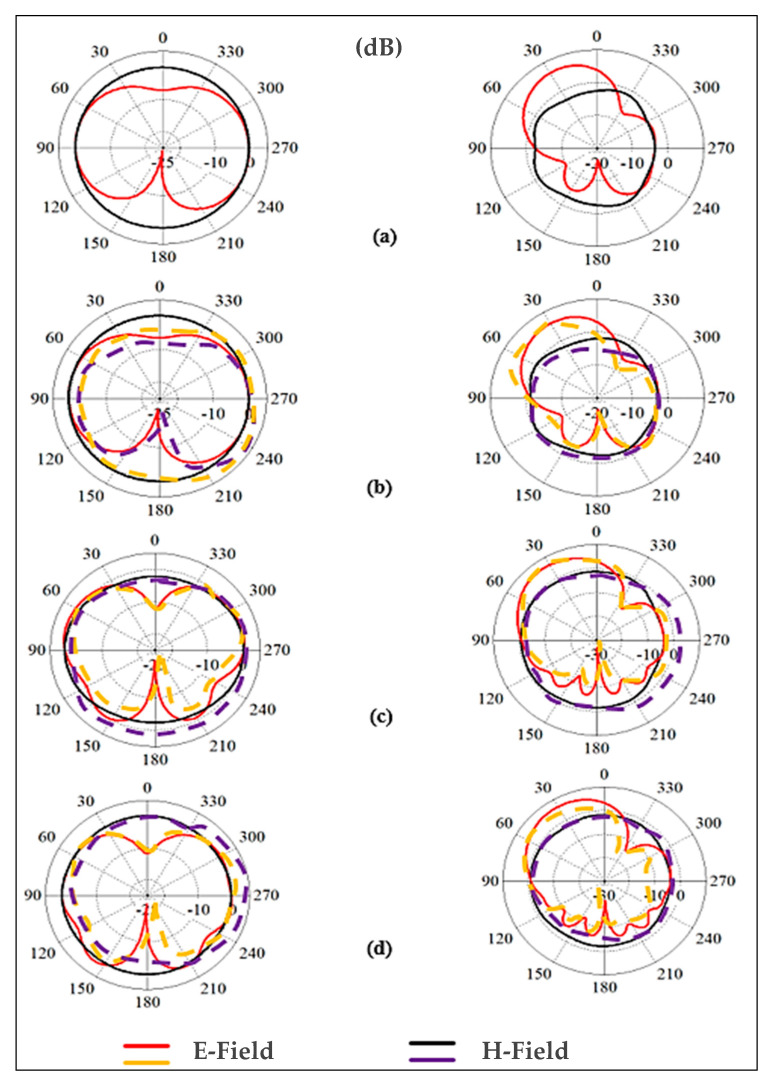
Simulated radiation patterns at 2.45 (left side) and 5.6 GHz (right side): (**a**) free space, no bending, (**b**) free space after bending, (**c**) bent on the layered chest, and (**d**) bent on the cylindrical arm. E field: yellow/red (normal/large sizes, left (2.45 band) and right sides (5.6 GHz), and H-field: black/purple) (normal/large sizes, left (2.45 GHz) and right sides (5.6 GHz)).

**Figure 19 micromachines-13-00475-f019:**
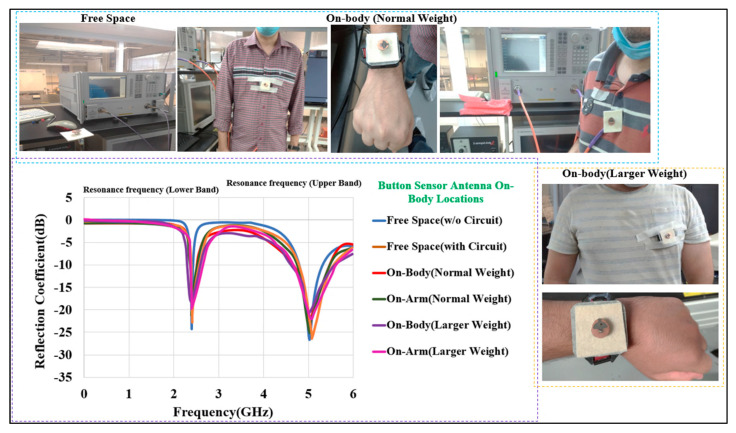
Measured S_11_ of the BSA in free space and on-body (chest and small and large size arm diameter of 50 mm and 100 mm, respectively).

**Figure 20 micromachines-13-00475-f020:**
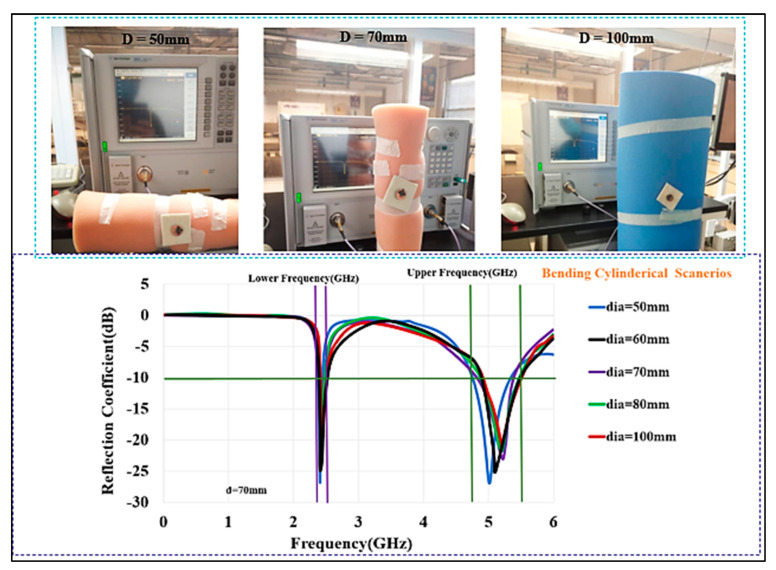
Measured S_11_ of the BSA under bending with different cylindrical curvatures (D (dia, in mm): 50, 60, 70, 80, and 100).

**Figure 21 micromachines-13-00475-f021:**
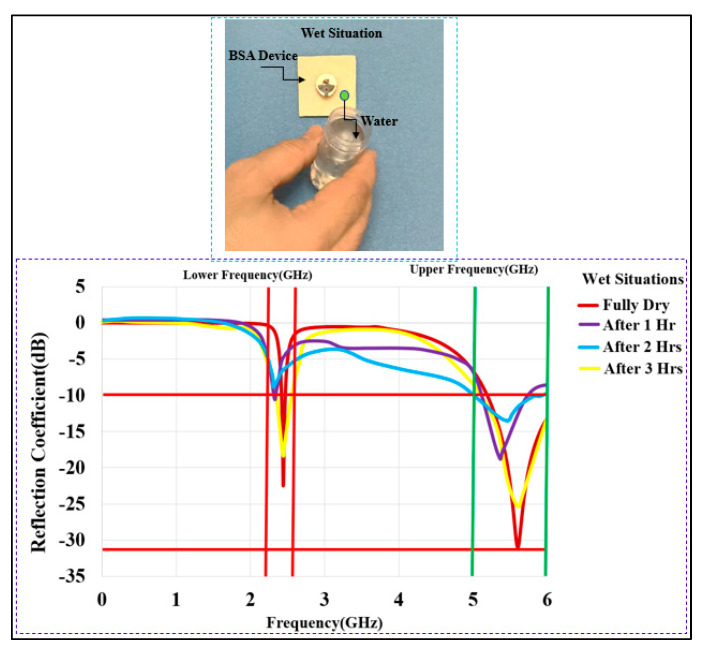
Measured S_11_ of the BSA in wet situations (dry, after 1 h (1 Hr), 2 h (2 Hrs), and 3 h (3 Hrs)).

**Figure 22 micromachines-13-00475-f022:**
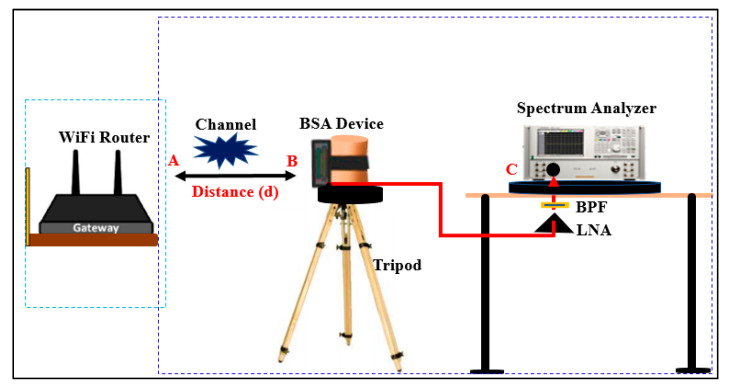
Set-up for the path loss. A: WiFi router, B: BSA, C: spectrum analyzer, BPF: band pass filter, LNA: low noise amplifier.

**Figure 23 micromachines-13-00475-f023:**
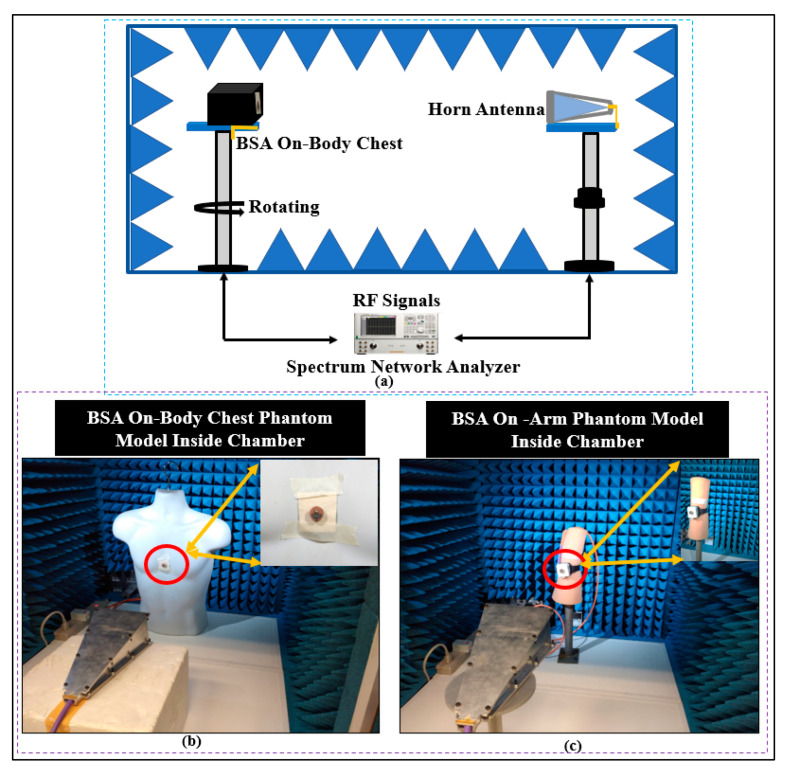
(**a**) Setup for far-field measurement, (**b**) phantom for the chest position, and (**c**) phantom for the arm position.

**Figure 24 micromachines-13-00475-f024:**
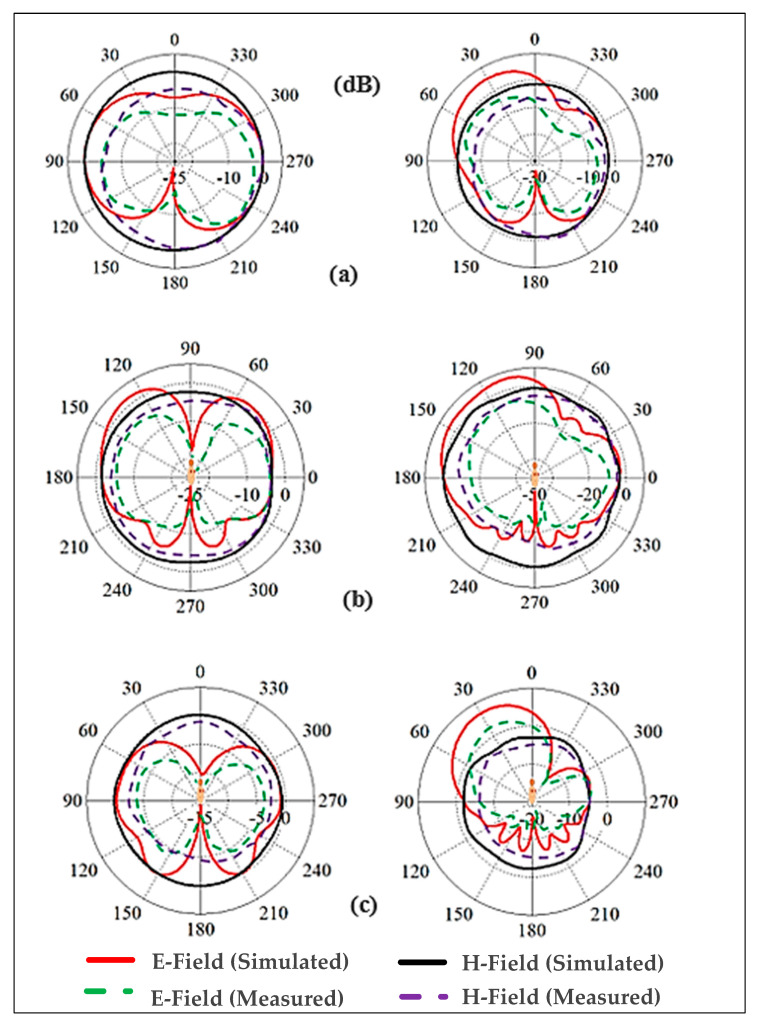
Simulated and measured radiation patterns of E- and H-planes. Left side: the lower frequency band (2.45 GHz), right side: the higher frequency band (5.6 GHz), (**a**) free space, (**b**) on-chest, (**c**) on-arm.

**Figure 25 micromachines-13-00475-f025:**
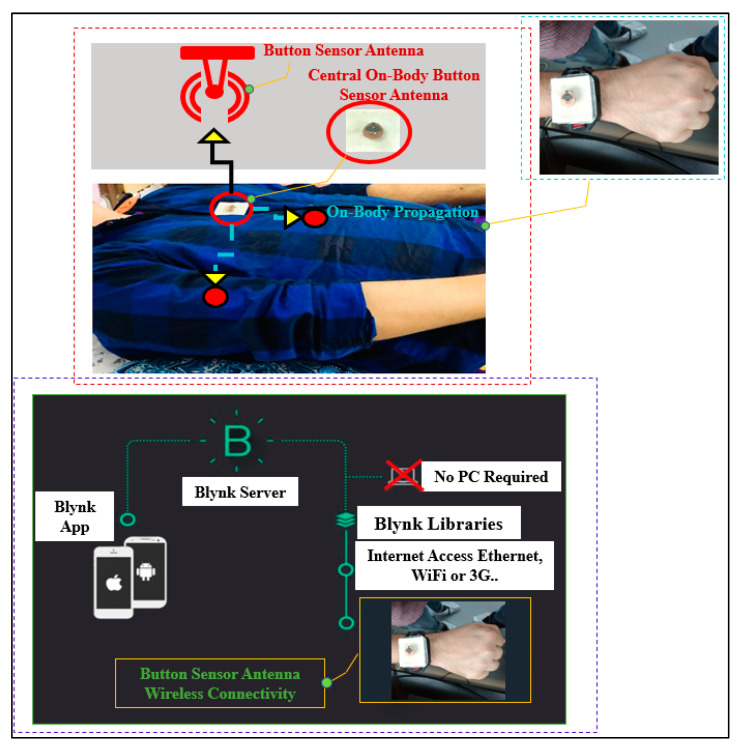
Measurement scenarios of wireless connectivity setup for on-body communication.

**Figure 26 micromachines-13-00475-f026:**
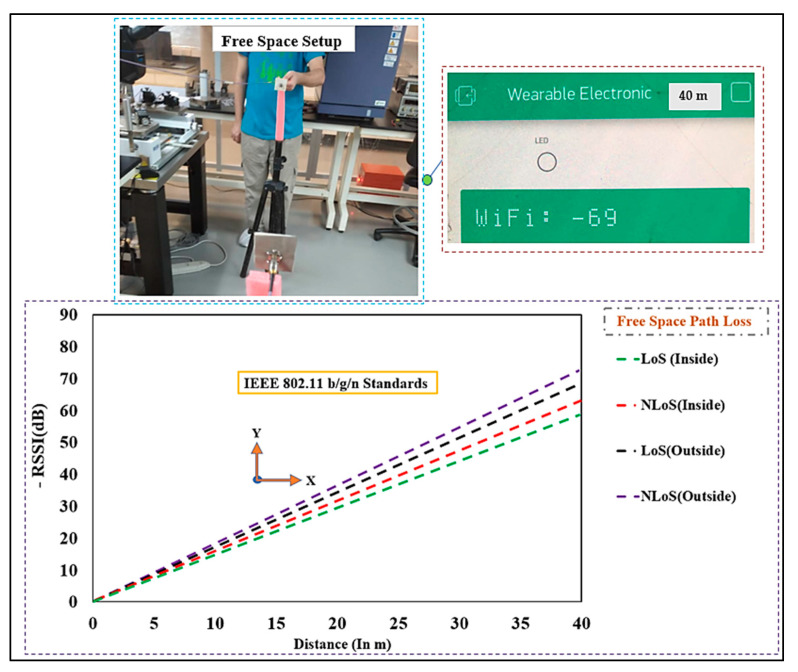
Measured received signal strength indicator (RSSI in dB) in free-space scenarios (Distance: 2 m to 40 m).

**Figure 27 micromachines-13-00475-f027:**
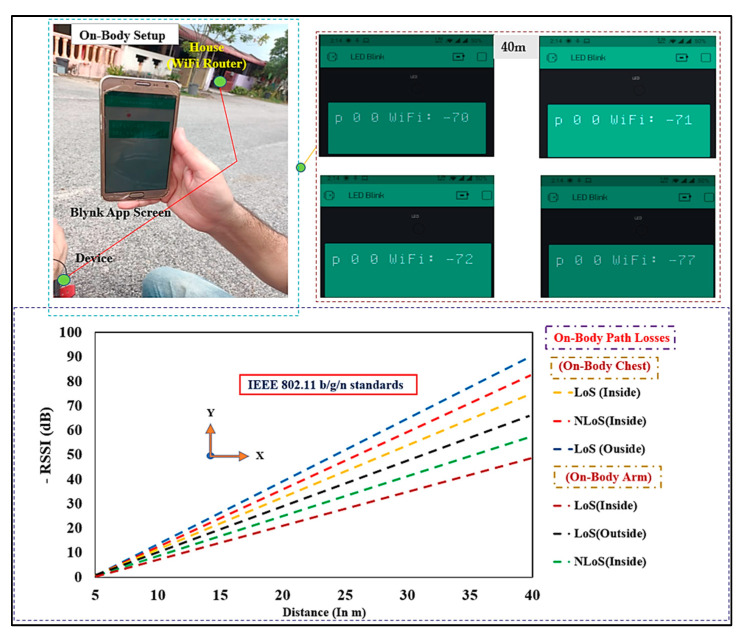
Measured RSSI in dB in on-body scenarios (Distance: 2 m to 40 m).

**Figure 28 micromachines-13-00475-f028:**
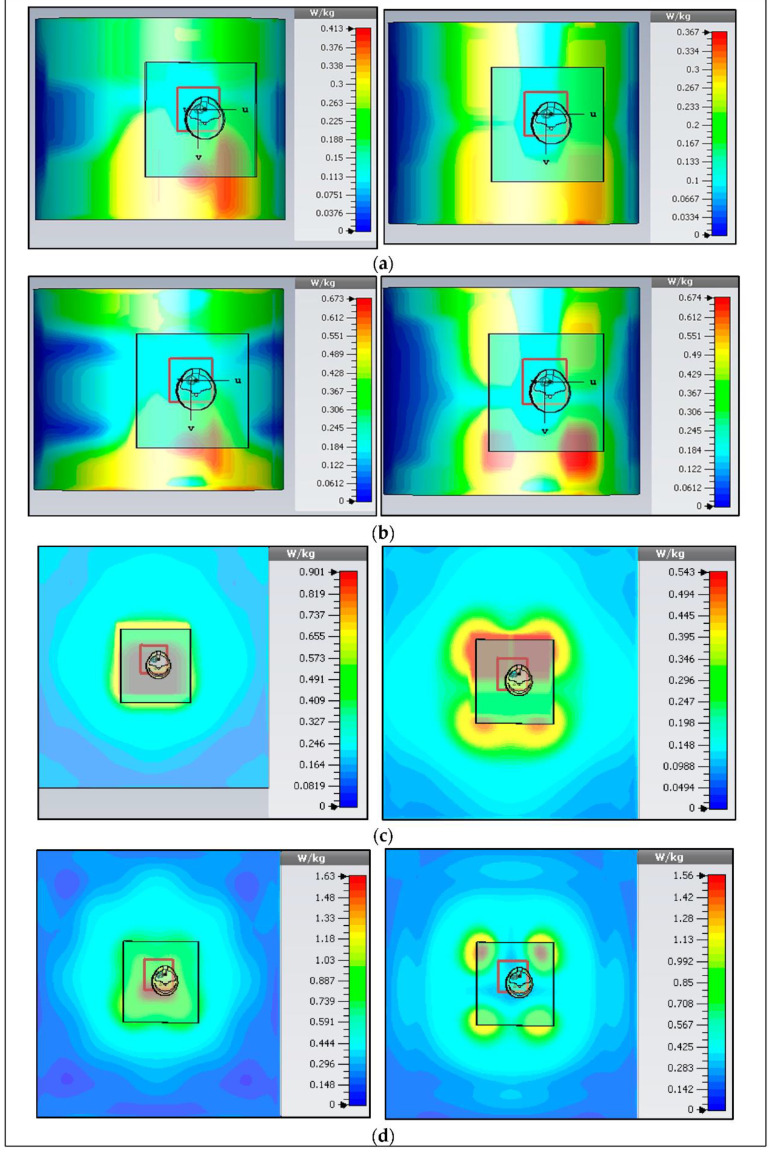
Simulated SAR (10 g, and 1 g) in the lower (upper) bands. (**a**,**b**) on-arm (at 2.38 GHz and 5.4 GHz), and (**c**,**d**) on-chest (at 2.38 GHz and 5.46 GHz)).

**Table 2 micromachines-13-00475-t002:** Summary of the Used Matching Network Components.

Matching Components	Value	Model Number
C_1_	1 pF	CAP0603-CAP
L_1_ and L_2_	1 nH	INDUCTOR 0603

**Table 3 micromachines-13-00475-t003:** Different SAR Values in Flat Positions.

Frequency (GHz)	1 g/10 g (On-Body)		1 g/10 g (On-Arm)		SAR Limits (1 g-US/10 g-EUR) 1.6/2.0 (W/Kg)
2.73/5.466	1.63/1.56	0.90/0.54	0.67/0.67	0.41/0.37	

Input power to the Button Sensor Antenna is 0.5 W or 100 dBm.

**Table 4 micromachines-13-00475-t004:** Different SAR Values in Bending Positions.

Frequency (GHz)	1 g/10 g (On-Body)		1 g/10 g (On-Arm)		SAR Limits (1 g-US/10 g-EUR) 1.6/2 (W/Kg)
2.41/5.49	0.89/1.55	0.57/0.64	0.95/1.46	0.54/0.73	

**Table 5 micromachines-13-00475-t005:** Summary Performance of the Proposed Button Sensor Antenna in Flat Position.

Characteristics	Button Sensor Antenna (Lower/Upper Band)		
Antenna Parameters	Free Space	On-Chest	On-Arm
Simulated S_11_ (dB) at the Center Frequency	−29.30/−30.97	−23.07/−27.07	−30.76/−31.12
Measured S_11_ (dB) at the Center Frequency	−27.45/−39.31	−21.35/−22.05	−30.15/−27.10
Simulated and Measured Center Frequency (GHz)	2.35–2.44/ 5.01–5.82	2.33–2.41/ 5.11–6.00	2.33–2.45/ 5.12–5.96
Voltage Standing Wave Ratio VSWA	1.08/1.01	1.31/1.16	1.17/1.11
Simulated Gain (dBi)	1.75/5.69	2.09/6.7	2.16/5.67
Directivity (dBi)	1.81/6.94	3.95/8.19	2.38/6.25
Simulated Radiation Efficiency (Total Efficiency)%	71.91/92.51 (67.17/92.15)	65.12/81.63 (59.26/80.0)	75.0/87.58 (50.0/75.64)
Radiation Patterns	Omnidirectional	Omnidirectional	Omnidirectional
(2 m/40 m) Path Loss (dB) Minimum/Maximum	34/82	35/84	34/83
Severe Bending and Crumpling Conditions/Wet Conditions	High Stability	Good Stability	Good Stability

**Table 6 micromachines-13-00475-t006:** Summary of the Design Performance Measured in Bending Situations.

Characteristics	Bending Scenario (Lower/Upper Band)		
Antenna Parameters	Free Space	On-Chest	On-Arm
Simulated S_11_ (dB) at the Center Frequency	−21.55/−21.96	−17.73/−35.00	−18.92/−28.11
Measured S_11_ (dB) at the Center Frequency	−20.31/−21.05	−17.25/−33.94	−17.85/−28.01
Simulated and Measured Center Frequency (GHz)	2.38−2.44/ 5.01−5.80	2.31−2.48/ 5.01−6.00	2.31−2.46/ 4.76−5.26
Simulated/Measured Bandwidth (MHz)	59.80/814.10	166.20/982.40	1261.70/524.20
Voltage Standing Wave Ratio VSWR	1.26/1.18	1.50/1.02	1.15/1.05
Simulated Gain (dBi)	1.35/6.09	1.31/6.35	0.91/6.45
Simulated Directivity (dBi)	1.78/6.34	2.8/7.14	2.31/7.19
Simulated Radiation Efficiency (Total Efficiency) %	75.00/95.00 (72.00/95.00)	71.00/82.98 (45.00/81.37)	73.00/85.00 (48.00/84.00)
Radiation Pattern Type	Omnidirectional	Omnidirectional	Omnidirectional
(10 m/40 m) Path Loss (dB) Minimum/Maximum	34/82 (LoS/NLoS)	35/83 (LoS/NLoS)	34/81 (LoS/NLoS)

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
