# Peer review of "Design and Evaluation of a Button Sensor Antenna for On-Body Monitoring Activity in Healthcare Applications"

_micromachines, 2022, doi:10.3390/mi13030475_

Round 1

Reviewer 1 Report

This paper presents a study on a dual-band button sensor antenna for on-body monitoring in the wireless body area networks systems. The topic is interesting and timely, the paper is very well-structured, with comprehensive studies with on-body simulations, measurements on the phantoms and volunteer. Antenna characteristics are evaluated extensively in different scenarios and conditions and the results are reported very clearly and consistently.

The buttom antenna seems to work well both at ISM 2.45 GHz and 5.8 GHz and is obviously a good candidate for health monitoring applications

I would give strong accept this paper due to very extensive studies in different scenarios and very well-structured paper. Especially I appreciate measurements both on phantoms and volunteer as comparison for simulations.

Only a few issues could be improved before publishing:

-literature review could be a bit more extensive including the newest flexible/textile on-body antenna references

- literature review could include discussion of advantages of the button antenna respect to the traditional flexible/textile antennas

-phantoms could be described more in detail, e.g. for which frequency range they are designed

- it would be good to emphasize novelty of this antenna respected to the previously presented studies also in conclusion section

-quality of some figure should be improved (at least Fig. 1 and 25)

Author Response

We thank the reviewers for the time spent on the review and the comments.

Reviewer 2 Report

This paper presented an antenna circuit on a mini PCB, which can be mounted onto human body with the help of a felt sheet. This paper did a systematic design and evaluation on the circuit, and results showed an acceptable communication feature of the circuit. However, I do not support its publication at the current form. Many works are needed to improve this paper. The following is my comments.

1.The novelty and importance of this works should be stated. The submitted work is only a circuit for wireless communication, the possible feature is its size and an additional substrate for human body. I’m doubting its novelty and importance. There have been many great works on the flexible and stretchable antennas for wearable electronics (e.g. the works by J.A. Rogers from more than 10 years ago). These antennas can even simultaneously sense different human motions and transmit the signals without external powers. When comparing with these works, the importance and novelty should be further discussed.         

2.The application should be improved. Only communication performance is tested. How about transmitting the signals from a sensor or monitoring some activities of human?

  1. About the model of chest and arm. The influence of fat and muscle may be discussed, but the miss of bone layer and the improper dimension for fat, muscle make the results not very credible.
  2. The figures have a too low resolution, and many information is not readable. The source of pictures in Figure 1 should be provided or cited.
  3. The English writing should be improved.

Author Response

(The authors gave the same response as above.)

Round 2

Reviewer 2 Report

Thank you for taking my suggestions, and I’ve get some promotions from the revised version. However, some misunderstanding happens when the authors describing the flexible devices reported by researchers (e.g. Pro. JA Rogers). The epidermal devices are also in the patch form, not directly painting elements on skin. Compared with these devices, the authors should further distinguish their works, maybe the cost, fabrication methods, available IC chips and transmittability. So, an improved description about the novelty and superiority is suggested in introduction.

Author Response

(The authors gave the same response as above.)
